# Nanotechnology to Improve the Performances of Hydrodynamic Surfaces

**Ali Alshehri [1,\*], Philippe Champagne [2], Laurent Keirsbulck [3] and El Hadj Dogheche [1]**

[1] IEMN-DOAE CNRS, Université Polytechnique Hauts-de-France, F-59313 Valenciennes, France;
elhadj.dogheche@uphf.fr

[2] LMCPA, Université Polytechnique Hauts-de-France, F-59600 Maubeuge, France;
philippe.champagne@uphf.fr

[3] LAMIH CNRS, Université Polytechnique Hauts-de-France, F-59313 Valenciennes, France;
laurent.keirsbulck@uphf.fr

\* Correspondence: ali.alshehri@etu.uphf.fr; Tel.: +33-6-31-14-30-87

**Abstract:** Nature continues to inspire scientists to adapt solutions in order to satisfy human needs, mainly in the maritime domain with metallic surface corrosion and its mechanical friction. In this research, the source of innovation comes from the lotus leaf and its well-known super-hydrophobicity. In this study, we have investigated the lotus leaf as a model for a super-hydrophobic maritime surface. The hydrothermal technique, which is considered to be a simple, low-cost, and scalable coating method, is applied to create zinc oxide (ZnO) nanorods (NRs), and an evaporation method is used to apply octadecyltrimethoxysilane (ODS). We apply such eco-green coatings onto commercial epoxy paints. Superhydrophobic surfaces (SHS) are obtained on maritime aluminum substrates. The characterization of SHS indicates improved behavior of water droplets on the treated surface: higher water static contact angles (WCA) from 98° to more than 152° and reduced sliding angles (SA) from 46° to 7°. Sliding speeds (SS) have been largely raised from 54 in the epoxy case to 1300 mm·s$^{-1}$ after treatment. These results clearly demonstrate the real opportunity to apply ZnO-based nanomaterials onto existing commercial maritime coatings.

**Keywords:** superhydrophobic surfaces; nanotechnology; nanorods; ZnO; water contact angle; sliding angle; sliding speed; hydrodynamic surfaces; epoxy paint

## 1. Introduction

Lotus leaf has become a reference model for super-hydrophobicity and self-cleaning surfaces with better stability and water repellency. The relevant properties are related to the micro and nanostructure, the chemical composition of the waxes, and its mechanical property with its competitors. The extraordinary shape and the density of the papillae are the basis for the extremely reduced contact area between surface and waterdrops. The exceptional, dense layer of very small epicuticular wax tubules is a result of their unique chemical composition. The mechanical robustness of papillae and wax tubules reduce damage and are the basis for the improvement and durability of water repellency [1,2]. Previous works in the literature report the fabrication of superhydrophobic coatings dedicated to ship hulls [3–5].

The different surface reactions against deposed waterdrops are known as the surface hydrophobicity or wettability. This is a very interesting subject for scientists to study. Young in 1805 [1] described the water contact angle (θ) as a relation of surface energy (γ) interaction between the 3 contacting surfaces—liquid (l), solid (s), and gas (g)—when a drop of water is laid on a solid surface

$$\cos(\theta_y) = \frac{\gamma_{sg} - \gamma_{sl}}{\gamma_{lg}} \qquad (1)$$

$\gamma_{sg}$ is the surface energy at solid-gas interface, $\gamma_{sl}$ is the surface energy at solid-liquid interface, and $\theta_y$ is water contact angle from Young equation.

Wenzel in 1936 [6] introduced the surface roughness ($R_a$) effect on the Young Equation (1):

$$\cos(\theta_w) = R_a \cos(\theta_y) \text{ with } \theta_w \text{ is the angle in the Wenzel regime} \tag{2}$$

Furthermore, in 1944, Cassie and Baxter [7] introduced the area fraction ($f$) in the Wenzel model by studying the effect of trapped air pockets in porous surfaces:

$$\cos(\theta_{CB}) = f(\cos\theta_y + 1) - 1 \text{ with} \\ \theta_{CB} \text{ is the angle in the Cassy Baxter regime} \tag{3}$$

It is well known that zinc oxide (ZnO) nanomaterials are frequently employed for their antibacterial capacities in a very wide field of applications [8,9]. The antimicrobial capacity of ZnO nanorod (NR) material has also been confirmed in paint [10]. In other applications, such as textile [9], the embedment of ZnO NRs in the epoxy coat has improved anticorrosion and hydrophobic properties with a water contact angle (WCA) of 128° [11]. An epoxy coat blended with ZnO nanorods has proven to provide very good corrosion resistance capacity [12–15]. The addition of ZnO nanoparticles (NPs) in epoxy paint improves the adhesion and the lifetime of the coat [14]. In our study, we investigate the super-hydrophobicity of treated ZnO NRs and the dynamic gain in the sliding speed on an epoxy-painted maritime aluminum disc.

## 2. Materials and Methods

### 2.1. Paint Application

We used an aluminum disc of 100 mm diameter as the substrate—a maritime application aluminum alloy 6061 from Goodfellow (Lille, France). Furthermore, this aluminum substrate was coated with an epoxy paint applied by Damen Shipyards Dunkerque (Dunkerque, France), using two layers of commercial paints by means of airless spray. A primer layer of 40 μm thickness (Intergard 269, International) was used to prepare the surface before a 250 μm thickness epoxy layer (Interzone 954, International) was applied. A curing period of 5 days in an atmospheric temperature was observed before the next step.

### 2.2. SuperHydrophobic Surface Creation

In order to test our solution on an industrial coat, we deployed two steps to employ the ZnO NRs on the painted substrate via a hydrothermal method; the first step was deployed to place the ZnO seeds, and the second step was to grow the ZnO NRs.

#### 2.2.1. ZnO Seed Synthesis

The seeds were prepared with zinc acetate dihydrate (Fisher, Hampton, NH, USA) and granulated sodium hydroxide (98%, Alfa Aesar, Haverhill, MA, USA) in absolute methanol (Fisher) as follow:

- Solution A was obtained by dissolving 90 mmol/L of zinc acetate dihydrate in methanol.
- Solution B was obtained by dissolving 75 mmol/L of sodium hydroxide in methanol.
- Solution C was obtained by slowly adding solution B (1 drop/s) to the solution A while stirring at 60 °C. At the end, a transparent seed solution was obtained.

After a stirring treatment (3 h at room temperature), the seed solution C was used for the preparation of the sample. The sample was immersed in the solution during the stirring process (duration 5 min). Then, the sample was dried at 90 °C for 20 min. This process was repeated five times to ensure uniform seed application onto the sample. Finally, the sample was annealed at 90 °C for 30 min. Figure 1 illustrates the seed preparation protocol.

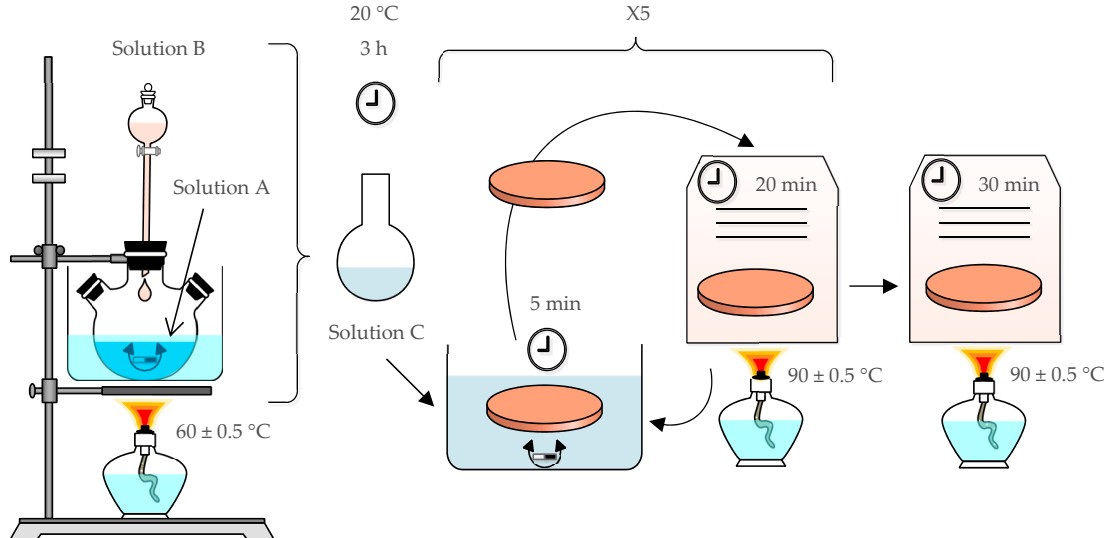

**Figure 1.** Seed preparation process.

### 2.2.2. ZnO Nanorod Synthesis

In the second step, nanorods were grown via a hydrothermal method by adding 100 mmol/L of zinc nitrate hexahydrate (Extra Pure, SLR, Fisher) and 100 mmol/L hexamethylenetetramine (ACS, 99+%, Alfa Aesar) solutions. A total of 250 mL of each solution was added to a Teflon bottle containing the sample. The mixture was agitated at 90 °C for 24 h. Then, the samples were washed in distilled water three times and dried at 90 °C for 30 min. Figure 2 shows the ZnO nanorod synthesis process.

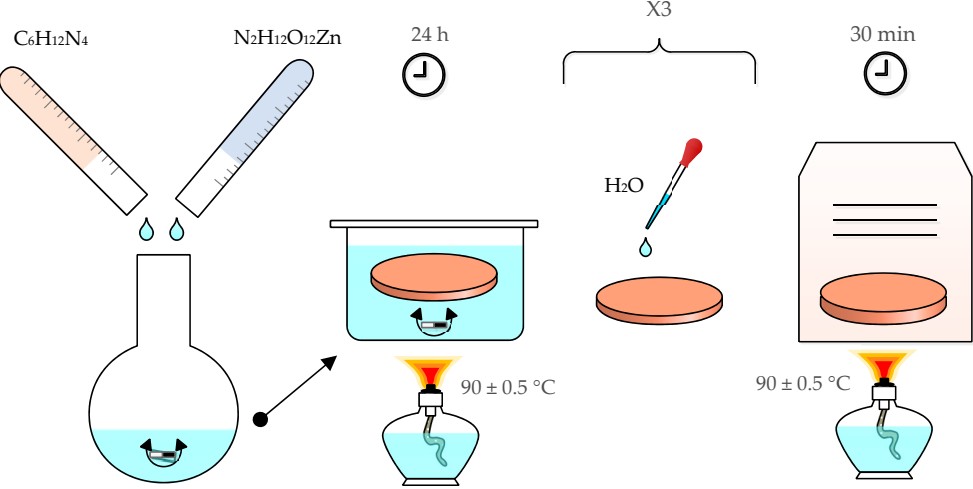

**Figure 2.** Zinc oxide (ZnO) nanorod preparation process.

### 2.2.3. Octadecyltrimethoxysilane (ODS) Deposition

A lower energy layer is necessary above the ZnO NRs. This layer was developed via an evaporation method. In brief, 200 μL of octadecyltrimethoxysilane (90%, Tech., ACROS Organics, Geel, Belgium) was placed in the bottom of a Teflon bottle, and the sample was held horizontally for 6 h at 90 °C.

## 3. Characterizations

### 3.1. Morphology Properties

The microscopic analysis of the ZnO NRs raised on the epoxy-painted aluminum disc was done via scanning electronic microscopy (SEM, ULTRA 55VP, ZEISS, Oberkochen, Germany). The surface morphology observations were done by microscope confocal (DCM 3D, Leica, Wetzlar, Germany).

### 3.2. Hydrophobic Properties

The sample was characterized to determine the improvement in the water contact angle and the sliding angle (SA) values of the three surface states:

- Aluminum substrate only;
- The substrate coated by the epoxy paint;
- The epoxy painted sample treated by ZnO + ODS.

WCA and SA were measured by the means of a drop shape analyzer (DSA25S, Kruss, Hamburg, Germany). A drop of distillate water of 2 μL was posed on the surface with a debit of 2.67 μL/s. The drop was then observed by using an integrated camera (TIS DFK 37BUX273, The Imaging Source, Bremen, Germany) to trace the surface line and the drop shape line. A Young Laplace analysis model was selected to determine the WCA. A total of 10 measurements were performed for the water angle contact (WCA) experiments on each type of surface. The SA was measured by the same system employing an automated tilt table with an inclining speed of 0.5°/s. Then, 10 measurements of sliding angles (SA) were also performed experiments on each type of surface. We also conducted contact angle hysteresis (CAH) measurements; the surfaces were tilted from 0° to 45°. A charge coupled device (CCD) digital video camera was used (25–30 images·s$^{-1}$) to record a movie of the sliding experiments. Once the droplets started to roll off the surface, we measured the advancing and receding contact angles. The difference between the advancing and receding contact angles corresponded to the contact angle hysteresis.

### 3.3. Hydrodynamic Properties

To measure the sliding speed (SS), the aluminum disc was held vertically, and a drop of distillate water was released freely on the top of the vertical disc side. The droplet movement was tracked by a high-speed camera (M310, Phantom, Wayne, NJ, USA) fitted with a 105 mm objective (AF-S VR Micro-Nikkor, Nikon, Osaka, Japan), and Phantom MultiCam Application software (version 675) was used to control the drops' travel distance and to find out the frequencies of the pictures. Indeed, 1000 pictures per second were taken and treated to determine the drop sliding speed. The measurements were done between each two points, respecting a constant traveled distance (d = 10 mm). To determine the time between two points, we converted their pictures frequencies to time (*t*):

$$t(\text{s}) = \frac{1}{1000} \times \left[ \frac{1}{fq(t)} - \frac{1}{fq(t + \Delta t)} \right] \qquad (4)$$

To control the drop size and flow, a syringe pump (100, Legato, KR Analytical, Cheshire, UK) was used to deliver a regular waterdrop of 50 μL with 1 mL/min. We note that 10 trials were performed for each surface type. Figure 3 shows the SS measurement protocol and two positions of a drop at two different moments.

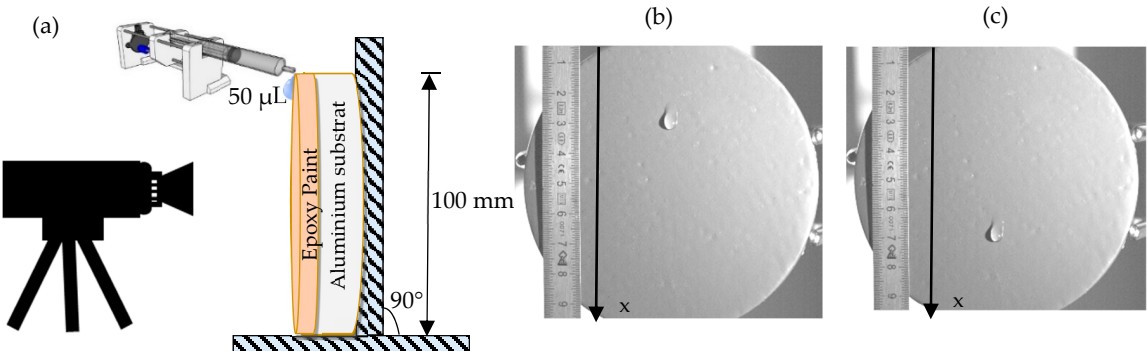

**Figure 3.** (**a**) Experimental setup for sliding speed measurement; (**b**) waterdrop image for the moment (*t*); (**c**) waterdrop image for the moment (*t* + Δ*t*).

## 4. Results and Discussion

### 4.1. Surface Morphology

The application of different layers on the aluminum substrate impacts the surface morphology. In our case, the surface roughness ($R_a$) of the nontreated aluminum substrate equaled 40 nm. However, the application of the epoxy paint reduced it to 26 nm. Finally, the application of our ZnO + ODS solution raised it to 58 nm. This variation of surface roughness illustrated by Figures 4–7 corresponds to the different steps of evolution of treated alumina substrate targeting a lotus leaf nanostructure.

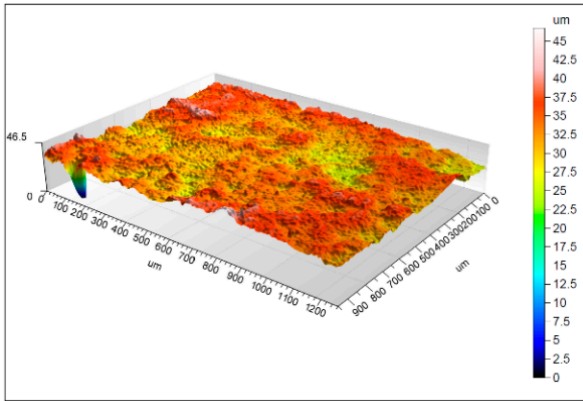

**Figure 4.** Aluminum surface morphology. Surface roughness ($R_a$) = 40 nm morphology.

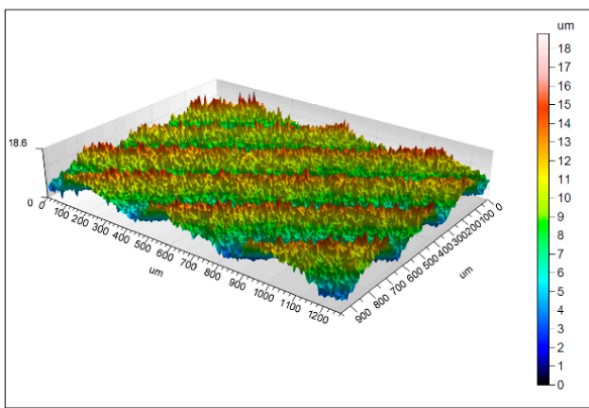

**Figure 5.** Aluminum surface + epoxy paint morphology. $R_a$ = 26 nm.

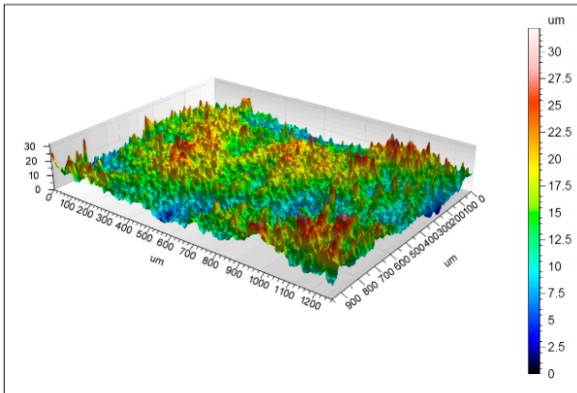

**Figure 6.** ZnO + octadecyltrimethoxysilane (ODS) surface morphology. $R_a$ = 58 nm.

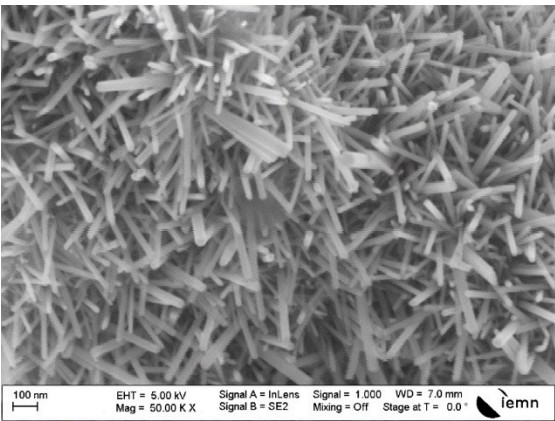

**Figure 7.** SEM analysis for ZnO nanorods deposited on aluminum substrate coated by epoxy paint.

### 4.2. Nanostructural Results

The obtained ZnO NRs on the epoxy layer after SEM analysis are shown in Figure 7. NRs nanostructure has a vertically orientation with an average length of 300 nm and a diameter of 30 nm. Obviously, the nanorods distribution is very dense with a great covering rate. This nanostructure helps to reduce the surface contact between waterdrops and the treated surface in order to create SHS beside the low-energy layer of ODS [16].

### 4.3. Wetting Results

Wetting properties of our proposed ZnO-based nanostructure were examined by contact angle measurements in static and hysteresis modes. While WCA and SA values provide respectable indication of the surface wetting state, for WCA < 90° and SA > 10°, the surface was considered to be hydrophilic. However, the surface was labeled hydrophobic when it had WCA > 90° and SA < 10°. Furthermore, a superhydrophobic surface was attributed to the surfaces with WCA > 150° and SA < 10°. According to the experimental results on our different surface states, hydrophilic surfaces were obviously observed with the aluminum surface and the epoxy-painted aluminum one. However, a SHS only appeared after the ZnO + ODS treatment. The contact angle hysteresis is corresponding to the wetting behavior of our ZnO + ODS-treated samples: We extracted the contact angle hysteresis around 5° for a liquid surface energy higher than 60 mN/m. Table 1 summarizes the WCA and SA through the three stages of our treated aluminum surface. Regarding the reliability aspects, many authors have largely investigated the mechanical damage for superhydrophobic coatings in commercial maritime applications [17]. Excellent durability of nanorod-functionalized samples has been verified in a previous study through washing and abrasion tests [18].

**Table 1.** Water contact angle (WCA) and sliding angle (SA) of the aluminum substrate of the three different surface states; (a) nontreated aluminum surface, (b) epoxy-painted surface, (c) epoxy-painted surface treated with ZnO + ODS, (d) our target, the superhydrophobic surface (SHS), and (e) lotus leaf measurements [16].

| Substrate | WCA (°) | SA (°) |
|---|---|---|
| (a) Aluminum | 93 | 22 |
| (b) Epoxy | 97 | 46 |
| (c) ZnO + ODS | 152 | 7 |
| (d) SHS | 150 | 10 |
| (e) LL | 153 | 26 |

*4.4. Hydrodynamic Results*

The sliding speed (SS) measurement of the free vertical slipping drops on the aluminum disc shows the behavior of the water droplet on the aluminum disc. Figure 8 shows the SS of a 50 µL water droplet traveling a 100 mm distance on the three different surfaces. At the beginning, we observed low-speed movement of water droplets on the nontreated aluminum substrate ($SS_{max}$ = 249 mm·s$^{-1}$). Coating the aluminum substrate with an epoxy paint will lower speed ($SS_{max}$ = 54 mm·s$^{-1}$). However, we observed an important gain in the SS after applying the ZnO + ODS treatment on our epoxy-painted aluminum substrate with a $SS_{max}$ of 1300 mm·s$^{-1}$. This observed speed was even greater than the sliding speed on the lotus leaf ($SS_{LL}$ = 319 mm·s$^{-1}$) [16].

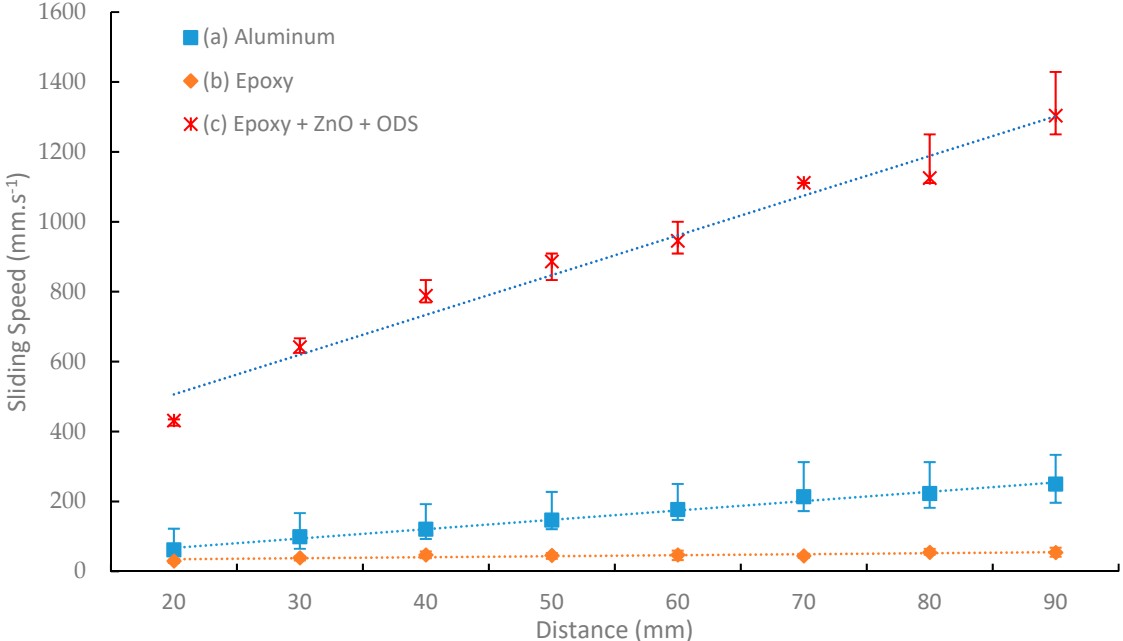

**Figure 8.** Sliding speed (SS) measurement for three surface stages: (a) nontreated aluminum surface; (b) epoxy-painted surface; (c) epoxy-painted surface treated by ZnO + ODS.

## 5. Conclusions

Nanorods were utilized successfully on an aluminum disc precoated with an industrial epoxy paint; the geometry of ZnO NRs is 300 nm in length and 30 nm in diameter. The obtained NRs were covered by an ODS top layer in order to attain a superhydrophobic surface. Ultimately, a SHS was obtained and characterized. A WCA of more than 152°, a SA of less than 7°, and a CAH close to 5° were obtained. The surface roughness was also raised from 26 in the epoxy case to 58 nm after treatment. The SS progress was also investigated on the three different surface states: aluminum, painted aluminum, and ZnO NR-treated painted aluminum. The SS raised considerably from 54 in the epoxy-painted surface to more than 1300 mm·s$^{-1}$ in the ZnO NR-treated one (lotus leaf: SS = 319 mm·s$^{-1}$). These findings can be adapted to maritime surfaces and can help to enhance hydrodynamic performance to reduce mechanical friction and fuel consumption.

**Author Contributions:** For research articles with several authors, E.H.D. contributed to the project management, P.C. and A.A. contributed to the synthesis of nanomaterials, L.K. to the hydrodynamic experiments.

**Funding:** This research was financially supported by Saudi Cultural Bureau in Paris.

**Acknowledgments:** We would like to acknowledge Christophe Callewaert, Laurent Rembotte, Hubert Vincent, and Damen Shipyards Dunkerque for their participation in the substrate painting. We also wish to thank Sirehna Naval Group Nantes for the helpful comments and discussions. This study was financially supported by Saudi Cultural Bureau in Paris.

**Conflicts of Interest:** The authors declare no conflicts of interest.

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
