# Peer review of "Nanotechnology to Improve the Performances of Hydrodynamic Surfaces"

_coatings, doi:10.3390/coatings9120808_

Round 1

Reviewer 1 Report

This manuscript is well written and presented nicely with data. Authors may want to  work on the following comments to improve the manuscript:

1. But introduction should be revised to establish the importance/merit of the work. Also need to establish the significance of this works with respect to already published works in this area.

2. As most the superhydrophobic surfaces are delicate and fragile in nature. One important aspect/property of this surface need to evaluate if plan to use  in maritime application as mentioned by the author, would be mechanical durability of the surface. For example the mechanical durability can be checked by abrasion test as performed in literature (ACS Appl. Mater. Interfaces 2011, 3, 9, 3508-3514; https://doi.org/10.1021/am200741f). 

Author Response

Reviewer 1:

This manuscript is well written and presented nicely with data. Authors may want to work on the following comments to improve the manuscript:

1. But introduction should be revised to establish the importance/merit of the work. Also need to establish the significance of this works with respect to already published works in this area.

Introduction has been revised according reviewer comments.

2. As most the superhydrophobic surfaces are delicate and fragile in nature. One important aspect/property of this surface need to evaluate if plan to use  in maritime application as mentioned by the author, would be mechanical durability of the surface. For example the mechanical durability can be checked by abrasion test as performed in literature (ACS Appl. Mater. Interfaces 2011, 3, 9, 3508-3514; https://doi.org/10.1021/am200741f). 

The introduction has been modified in order to show the importance of this work regarding the existing literature.

Mechanical durability of ZnO synthetized by hydrothermal technique, has been already investigated in a previous study [reference 11]: taking account this results, we have focused mainly on other parameters as the hydrodynamic performances, etc…

Reviewer 2 Report

The authors present a low-cost technique for fabricating a superhydrophobic layer on an epoxy paint. The above method could be possibly used to reduce the drag of a ship‟s hull. The above topic is an important concern for the shipping industry, however, major revisions should be performed in the manuscript. In particular, the following issues have to be addressed by the authors:

1. The introduction section should be extended in order to include more references and information regarding previous works on fabricating superhydrophobic coatings for a ship hull.

2. Contact angle hysteresis measurements (advancing and receding contact angles) should be performed. Note that, apart from the high apparent contact angle, a small contact angle hysteresis is an essential requirement for a surface to be superhydrophobic.

3. The mechanical strength of the micro-structure, when subjected to water pressure, should also be discussed. Can the nanorods withstand the pressure of the surrounding water at the bottom of the ship's hull for long periods? (note that turbulence phenomena may also occur, increasing the overall pressure)

Author Response

Reviewer 2:

The authors present a low-cost technique for fabricating a superhydrophobic layer on an epoxy paint. The above method could be possibly used to reduce the drag of a ship‟s hull. The above topic is an important concern for the shipping industry, however, major revisions should be performed in the manuscript. In particular, the following issues have to be addressed by the authors:

1. The introduction section should be extended in order to include more references and information regarding previous works on fabricating superhydrophobic coatings for a ship hull.

I have included some new references in the manuscript

[3]  H. Dong, M. Cheng, Y. Zhang,  H. Wei and  F. Shi , Extraordinary drag-reducing effect of a superhydrophobic coating on a macroscopic model ship at high speed, J. Mater. Chem. A, 2013,1, 5886-5891

[4]  J. Simpson, S. Hunter and T. Aytug, Superhydrophobic materials and coatings: A review, Reports on progress in physics, 78 (2015) 08 6501

2. Contact angle hysteresis measurements (advancing and receding contact angles) should be performed. Note that, apart from the high apparent contact angle, a small contact angle hysteresis is an essential requirement for a surface to be superhydrophobic.

First, the contact angle hysteresis has been measured using KRUSS DSA system, this set-up is available in the lab and can perform static contact angle and contact angle hysteresis modes. In the study, we have mainly concentrated our experiments on the static mode on the prepared surfaces (WCA and SA). Regarding the reviewer comments, we have performed the contact angle hysteresis (CAH) as well. The contact angle hysteresis CAH is found around 5° for a liquid surface energy higher than 60mN/m.

 3. The mechanical strength of the micro-structure, when subjected to water pressure, should also be discussed. Can the nanorods withstand the pressure of the surrounding water at the bottom of the ship's hull for long periods? (note that turbulence phenomena may also occur, increasing the overall pressure)

This research represents an initial step of application of nanomaterials dedicated to metallic maritime substrates. We want to highlight the principle of nanostructured surface for shipping field. Sure, the pressure will affect the microstructure and the next step will be to propose alternative solution, more durable for a maritime application.

The global mechanical behavior will be investigated more deeply in a near future study. Turbulence phenomena and cavitation effect will be study through a collaboration with Prof Alicja Krella (Hydrodynamics Center, Institute of Fluid Flow Machinery, Polish Academy of Sciences)

Reviewer 3 Report

1. The superhydrophobicity of lotus leaf was mentioned as an inspiration for this research, the physical and chemical principle of this effect could be explained for better understanding the results.

2. I recommend incorporation of these newest references on ZnO nanorods:

Corrosion resistance of ZnO nanorod superhydrophobic coatings with rose petal effect or lotus leaf effect, DL Lai, G Kong, XC Li, CS Che - Journal of nanoscience

https://doi.org/10.1166/jnn.2019.16313

Hydrophobicity and Leidenfrost point of ZnO nanorod array combined with nanoscale roughness on the topmost surface

https://doi.org/10.1016/j.matchemphys.2018.06.075

Author Response

 Reviewer 3:

1. The superhydrophobicity of lotus leaf was mentioned as an inspiration for this research, the physical and chemical principle of this effect could be explained for better understanding the results.

Lotus leaf principle is basically explained in the introduction, as required by the reviewer.

2. I recommend incorporation of these newest references on ZnO nanorods:

I have included the proposed references in the manuscript.

[1] DL Lai, G Kong, XC Li, CS Che, " Corrosion resistance of ZnO nanorod superhydrophobic coatings with    rose petal effect or lotus leaf effect ", Journal of nanoscience, vol 19, n° 7, July 2019, pp. 3919-3928(10)

[2] T. Kano, T. Isobe, S. Matsushita and A. Nakajima, Hydrophobicity and Leidenfrost point of ZnO nanorod array combined with nanoscale roughness on the top most surface , Materials Chemistry and Physics, vol. 217, september 2018, pp. 192-198

Round 2

Reviewer 1 Report

I see some improvement in the introduction some general statement about "lotus effect" and cited reference to few the relevant studies. But I don't see much improve of the introduction part. Authors should details the previously published works in this area to establish the importance of this study. 

Reviewer 2 Report

I believe that the manuscript has been significantly improved and is now suitable for publication.